# HoBi-like Pestivirus Is Highly Prevalent in Cattle Herds in the Amazon Region (Northern Brazil)

**DOI:** 10.3390/v15020453

**Published:** 2023-02-06

**Authors:** Leticia F. Baumbach, Ana Cristina S. Mósena, Raquel S. Alves, Laura J. Camargo, Juliana C. Olegário, Leonardo R. Lobraico, João Marcos N. Costa, Mauro R. Borba, Fernando V. Bauermann, Matheus N. Weber, Cláudio W. Canal

**Affiliations:** 1Laboratório de Virologia, Faculdade de Veterinária, Universidade Federal do Rio Grande do Sul (UFRGS), Porto Alegre 90040-060, RS, Brazil; 2Secretaria de Defesa Agropecuária, Ministério da Agricultura, Pecuária e Abastecimento, Pedro Leopoldo 33600-000, MG, Brazil; 3Laboratório de Epidemiologia Veterinária, Faculdade de Veterinária, Universidade Federal do Rio Grande do Sul (UFRGS), Porto Alegre 90040-060, RS, Brazil; 4Department of Veterinary Pathobiology, College of Veterinary Medicine, Oklahoma State University (OSU), Stillwater, OK 74078, USA; 5Laboratório de Microbiologia Molecular, Instituto de Ciências da Saúde, Universidade Feevale, Novo Hamburgo 93510-235, RS, Brazil

**Keywords:** BVDV, phylogeny, serum neutralization, epidemiology

## Abstract

Pestiviruses are globally distributed and cause substantial economic losses to the cattle industry. In Brazil, the country with the world’s largest cattle population, pestivirus infections are well described in some regions, such as in the south, where a high frequency of BVDV-2 is described and contrasts with the high prevalence of HoBi-like pestivirus (HoBiPeV) in the northeast. However, there is a lack of information about pestiviruses in the Amazon Region, in northern Brazil, with a cattle population estimated at 55.7 million head, which has a significant impact on the international livestock market. Therefore, this study investigated the seroprevalence and genetic variability of ruminant pestiviruses in 944 bovine serum samples from four states in northern Brazil: Pará (PA), Amapá (AP), Roraima (RR), and Amazonas (AM). Our results showed that 45.4% of the samples were seropositive (19.8% for BVDV-1, 14.1% for BVDV-2, and 20.9% for HoBiPeV). All samples were tested by RT–qPCR, and three were positive and classified as HoBiPeV in a phylogenetic analysis. These serological and molecular results contrast with those from other regions of the world, suggesting that the northern Brazilian states have a high prevalence of all bovine pestiviruses including HoBiPeV.

## 1. Introduction

Bovine viral diarrhea virus (BVDV), genus *Pestivirus* and family *Flaviviridae*, is classified into the three pestivirus species that are most important to cattle health: BVDV-1 (*Pestivirus A*), BVDV-2 (*Pestivirus B*) and HoBi-like Pestivirus (HoBiPev-*Pestivirus H*) [1,2]. Bovine pestiviruses can also be classified into subgenotypes based on phylogenetic analysis of full-length genome, 5′UTR sequences, or N^pro^ and E2 analyses [3]. To date, several subgenotypes have already been described: BVDV-1 (1a–1x) [4,5,6], BVDV-2 (2a–2e) [7,8,9], and HoBiPeV (a–e) [10,11]. Moreover, BVDV can be classified into cytopathic (CP) and non-cytopathic (NCP) biotypes, according to their ability to induce cytopathic effect (CPE) during viral replication in cell culture. NCP strains represent the most field isolates. CP strains are isolated almost exclusively from cases of Mucosal Disease and originate from NCP strains through mutations and/or recombination in the genome [12].

Pestiviruses are single-stranded, positive-sense RNA viruses. They contain a viral genome of approximately 12.3 Kb, with a single open reading frame (ORF) flanked by two untranslated regions (UTR), 5′UTR and 3′UTR [13]. This ORF encodes a polyprotein that is cleaved into four structural proteins (C, E^rns^, E1 and E2) and at least seven nonstructural proteins (N^pro^, p7, NS2-3, NS4A, NS4B, NS5A and NS5B) [14,15]. Phylogenetic analysis usually uses sequences from the highly conserved 5′UTR. Additionally, N^pro^ and/or E2 regions are also frequently used for evolution, epidemiology, and taxonomy studies [16,17].

BVDV is globally distributed and tends to be endemic in regions with a cattle population [18]. BVDV infection can cause a wide variety of clinical manifestations, including enteric and respiratory disease and reproductive failure [18]. Pestiviruses have tropism for fetal tissues and can cause transplacental infections. In utero, BVDV infection can result in embryo-fetal death, stillbirth, abortion, and congenital defects (hydranencephaly, cerebellar hypoplasia, optic defects, arthrogryposis) depending on the stage of pregnancy at which infection occurred [12]. A key ruminant pestiviruses characteristic is that it can induce immunotolerance, resulting in persistently infected (PI) animals that continuously shed the virus throughout their lives, being the primary source of BVDV infection in herds [19].

Currently, BVDV it is considered one of the most important bovine pathogens worldwide and causes significant animal health impacts, leading to substantial economic losses to the cattle industry [3]. Furthermore, BVDV is part of the Bovine Respiratory Disease (BRD) complex, which contributes to increased susceptibility to other diseases. The BRD include other important viruses, such as bovine respiratory syncytial virus (BRSV; *Bovine orthopneumovirus*), bovine parainfluenza virus type 3 (BPIV-3; *Bovine respirovirus-3*) and bovine herpesvirus type 1 (BoHV-1) [20].

The most prevalent pestivirus species worldwide is BVDV-1 [3]. However, epidemiological studies show a worldwide distribution of BVDV-2, including in South America, Europe, USA and Japan [3,8,21]. More recently, in 2004, HoBiPeV was discovered and characterized from commercial fetal bovine serum (FBS) from Brazil [22]. Notably, HoBiPeV was identified in banked samples, dated 1996, from an outbreak of disease in Brazilian water buffalos, demonstrating the long-term circulation of the virus in the country [23]. In recent decades, several studies have described the presence of this emerging pestivirus, mainly in cattle from South America [24,25,26] and southeast Asia [27]. In 2011, HoBiPeV was described in an outbreak in southern Italy [28]; however, until now, it has not been reported in Europe anymore [29].

Bovine pestiviruses are widespread in Brazil and have high genetic variability [21,30]. In some Brazilian regions this variability is already well described, such as in the south [31] and northeast [24]. Almost 80% of the isolates that have been identified thus far were BVDV-1a and BVDV-2b, whereas BVDV-1b, 1d, 1e, and 2c were detected at low frequencies [21,32,33]. Despite an increasing number of reports of HoBiPeV in Brazil, epidemiological data from the northern region, also known as the Amazon region, are scarce.

Due to the economic importance of pestivirus infection, many countries have successfully eradicated these viruses [19,34,35]. These control efforts were based on the systematic identification and elimination of PI animals, biosecurity measures, and surveillance strategies [36]. Currently there is no official pestivirus control and eradication program in Brazil, despite the country holding the largest cattle population in the world with 224.6 million heads [37].

Understanding the genetic and antigenic variability of pestiviruses in different regions of Brazil is essential to support the development of effective control measures, accurate diagnostic tests, and effective vaccines. Brazil has continental dimensions, several biomes, and different animal husbandry systems. Northern Brazil has the second largest Brazilian herd, which has reached 55.7 million cattle heads, making the region extremely relevant in the regional and international cattle market. Moreover, the state of Pará occupies the third position in the ranking of Brazilian states, with the largest cattle herd, totaling 23.9 million heads [38]. Appendix A describes the number of cattle heads in different regions of Brazil. Toward this goal, this original study focused on determining the seroprevalence and genetic variability of pestiviruses in cattle in northern Brazil. Additionally, serological analysis was comparatively performed on strains of BVDV-1, BVDV-2 and HoBiPeV.

## 2. Materials and Methods

### 2.1. Target Population and Sample Size

Serum samples were collected in the federative states of Pará (PA), Amapá (AP), Roraima (RR), and Amazonas (AM) in northern Brazil (Figure 1). These samples were part of a surveillance study conducted by the Brazilian Department of Animal Health (DSA/SDA/MAPA) in 2014–2015 to obtain a foot-and-mouth disease (FMD)-free zone using vaccination status from the World Organization for Animal Health (WOAH). Animals between 6 and 24 months of age were randomly sampled in each state [39]. No data about immunization history, animal sex or the husbandry system (beef or dairy) are available. Serum samples were stored at −80 °C until analysis. A total of 944 cattle samples from the northern region of Brazil were recovered from the original study (PA—66 animals; AM—536 animals; AP—147 animals; RR—195 animals). The virus neutralization test was performed using a subsample of 390 bovine serum samples from northern Brazil (PA—29 animals; AM—226 animals; AP—57 animals; RR—78 animals). The sample number was calculated using online AusVet EpiTools Epidemiological Calculators (epitools.ausvet.com.au, accessed on 9 March 2020), and samples were selected through systematic random sampling [39].

### 2.2. RNA Isolation and RT–qPCR

A total of 944 serum samples collected in northern Brazil were tested by RT–qPCR. RNA was isolated using TRIzol LS reagent (Thermo Fisher Scientific, Waltham, MA, USA) according to the manufacturer’s instructions. Initially, all samples were tested in pools of approximately 24 samples each using the commercial kit RT–qPCR VetMAX-Gold bovine virus diarrhea RNA test (Applied Biosystems, Life Technologies, Austin, TX, USA). Then, samples from positive pools were tested individually by conventional RT–PCR. Complementary DNA (cDNA) synthesis and PCR were performed using GoScript™ Reverse Transcriptase and GoTaq^®^ DNA Polymerase according to the manufacturer’s protocol (Promega, Madison, WI, USA). Primers used to amplify fragments of the 5′UTR, N^pro^, and E2 pestivirus genomic region are described in Table 1. Amplification products were stained with Blue Green Loading Dye I (LGC Biotecnologia, Cotia, Brazil), subjected to electrophoresis in 1.5% agarose gel and visualized under ultraviolet light.

### 2.3. Sequencing and Phylogenetic Analysis

The genetic diversity of ruminant pestivirus species was verified by partial sequencing of the 5′UTR, N^pro^, and E2 genomic regions using RT–PCR-positive samples with the primers described in Table 1. The amplification products were purified using a PureLink PCR purification kit (Thermo Fisher Scientific) according to the manufacturer’s instructions. Both DNA strands were sequenced using Sanger methodology (ABI PRISM 3100 genetic analyzer; Big-Dye Terminator v.3.1 cycle sequencing kit; Thermo Fisher Scientific). The sequences were assembled with Geneious prime (v.2020.2.2; Biomatters Limited, Auckland, New Zealand), and a similarity search was performed using BLAST (nucleotide BLAST; https://blast.ncbi.nlm.nih.gov/Blast.cgi, accessed on 18 January 2022). For phylogenetic analysis, pestivirus reference sequences were retrieved from GenBank (https://www.ncbi.nlm.nih.gov/genbank/, accessed on 18 January 2022). Multiple alignments were performed using MUSCLE alignment [45]. Phylogenetic trees based on the 5′UTR, N^pro^, and E2 were performed by applying the maximum likelihood method (ML) [46] based on the general time reversible model with gamma-distributed plus invariant sites (GTR + G + I) [47]. All evolutionary analyses were conducted using the Molecular Evolutionary Genetics Analysis software package 6 (MEGA6) [48]. The robustness of the hypothesis was tested with 1000 nonparametric bootstrap analyses. The sequence data have been submitted to the GenBank database under accession numbers OP927798-P927805.

### 2.4. Virus Neutralization

The virus neutralization (VN) test is the gold standard method used to detect and quantify neutralizing antibodies against bovine pestiviruses in serum [49]. The VN assay was performed using three cytopathic (CP) strains: BVDV-1a (Oregon C24 V), BVDV-2a (SV-253), and HoBiPeV (Italy 83/10), hereafter referred to as BVDV-1, BVDV-2 and HoBiPeV strains.

Strains were propagated and titered in Madin–Darby bovine kidney cells (MDBK) grown in Dulbecco’s modified Eagle’s medium (DMEM; Thermo Fisher Scientific) supplemented with 100 units/mL penicillin, 100 μg/mL streptomycin, and 10% equine serum. MDBK cells (ATCC CCL-22) were previously found to be free of mycoplasmas and pestivirus RNA by RT–PCR [31,41,50].

The VN assay was conducted according to the World Organization for Animal Health [49]. Serum samples were heat-inactivated for 30 min at 56 °C. In a 96-well cell culture plate, triplicate samples were serially diluted (2-fold) in DMEM from 1:8 to 1:1024. Samples that neutralized the virus at a dilution of 1:1024 were further tested in dilutions from 1:512 to 1:32,768 [49]. Following dilution, 50 μL of virus solution containing the 100 median tissue culture infectious dose (TCID50) was added to each well. The plate was incubated for one hour at 37 °C, followed by the addition of 50 μL of MDBK cell suspension (1.5 × 10^5^ cells/mL). The plates were then incubated at 37 °C for 72–96 h in a 5% CO_2_ atmosphere.

The cells were examined for the presence of CPE under a light microscope, and samples were considered antibody-positive in the absence of CPE. The test was validated by a cell control, seronegative sample control, and back titration of the virus stock to check the viral potency. The resulting VN titers were calculated by Reed–Muench [51].

## 3. Results

### 3.1. RT–PCR and Phylogenetic Analyses

A total of 944 bovine serum samples were tested by RT–PCR. Samples from positive pools were tested individually using the primer pairs listed in Table 1 in individual reactions. Three samples (0.31%) tested positive for pestivirus. These samples were collected in the AM, AP, and RR states and named LV_510/20 AM, LV_195/20 AP, and LV_242/20 RR.

Upon nucleotide BLAST analysis (https://blast.ncbi.nlm.nih.gov/, accessed on 18 January 2022), the three sequences were found to share a high degree of identity with HoBiPeV strains: approximately 98 to 99% for the E2 region, 99% for N^pro^, and 100% in the 5′UTR. A phylogenetic tree was constructed for the partial 5′UTR, N^pro^, and E2 regions (Figure 2). The phylogeny of the 5′UTR for the sequence LV_510/20AM was not performed (Figure 2a), since RT-PCR resulted negative repetitively.

### 3.2. Virus Neutralization

Pestiviruses have great antigenic diversity. However, there is a serologic cross-reaction between the ruminant pestivirus species. Due to cross-reactivity, many times it is not possible to determine which species has the highest titer. To accurately detect antibody titers, OIE establishes a difference in titer by >4-fold to define a strain for which the antibodies have a specific neutralization reaction greater than the cross-neutralization reaction [49]. For cross-neutralization results within the 4-fold difference in titer between tested strains, serology at the species level is inconclusive and in this study they were named as “no predominant titer”. Samples with titers lower than eight were considered negative in this study. Antibody titers against BVDV-1, BVDV-2, and HoBiPeV are shown in Appendix A.

Comparative virus neutralization was performed with the BVDV-1, BVDV-2, and HoBiPeV reference strains tested against 390 bovine serum samples from northern Brazil. Of those, 177 samples were positive (seroprevalence of 45%, 95% confidence interval: 41-51%) (https://epitools.ausvet.com.au/ciproportion, accessed on 18 January 2022). Among the samples for which antibody specificity could be determined, 19.8% (35/177) had predominant titers against BVDV-1, 14.1% (25/177) against BVDV-2, and 20.9% (37/177) against HoBiPeV. Additionally, 45.2% (80/177) of the samples had a higher titer against one of the strains, although lower than the 4-fold threshold (Table 2). Among these 80 samples, variable combinations of positivity were identified (Figure 3).

## 4. Discussion

In countries free of foot and mouth disease virus, BVDV is considered the most important bovine virus and has been the target of numerous control and/or eradication programs [3,19]. Economic losses caused by pestiviruses cost millions of dollars a year, making BVDV one of the costliest viral diseases in the world [52]. Control programs should be based on the implementation of biosecurity measures aimed at preventing re-/introduction of the infection in free herds, the elimination of PI animals, and continuous surveillance [35]. The PI animals are a key element to control the spread of infection in the herd. The persistence of virus infection should be confirmed by resampling after an interval of at least 3 weeks. PI animals are usually seronegative [49].

In areas where the risk of introducing BVDV infection is high, one option is to implement vaccination, after removal of PI animals. The role of vaccines is as an additional biosecurity measure, increasing fetal protection and preventing the birth of PI animals [53]. The choice for including a vaccination regime may differ between countries/regions according to the genetic variability and prevalence of infected herds. The need to ensure vaccination adherence, the interference with the interpretation of serological test results, and safety and efficacy issues with the vaccines themselves must also be taken into account [54].

In Europe, more than 90% of circulating BVDV strains are BVDV-1 [3,4,17]. BVDV-1 is also predominant in Asia and the Americas. Conversely, BVDV-2 is widely reported in the USA, Brazil, and Japan [3]. Concerning genetic variability, BVDV subtype 1b is predominant worldwide, followed by BVDV-1a [3]. The genetic composition of pestiviruses in Brazil is diverse. Although BVDV-1a, BVDV-2b, and HoBiPeV represent almost 80% of the isolates identified thus far, the presence of the subtypes BVDV-1b, 1d, 1e, and 2c has been reported [21,24,31,32,33,55,56,57].

Despite its increasing detection in Brazilian herds, information on the prevalence and genetic variability of pestiviruses in some Brazilian regions is still scarce. Despite having a late introduction of livestock, the Amazonia region has a livestock density similar to that of traditional producing states and holds the second-largest Brazilian herd [37]. Brazil is the fifth largest exporter of live cattle globally and holds 8.5% of the world trade in live animals (Appendix A) [38]. This demand emerged to meet specific requirements of international markets, such as slaughter for religious beliefs in Middle Eastern countries. The Pará state is currently the largest Brazilian exporter of live cattle [58]. The Vila do Conde Port is Brazil’s main international departure point, with more than 90% of live cattle exports from Brazil [58]. Brazil’s principal markets for live cattle in recent years have been Turkey, Iraq, Lebanon, Egypt, Saudi Arabia, and Jordan [59]. The northern cattle herd is increasing more than the rest of the country’s average [37].

In our study, we detected active pestivirus infection by RT–PCR in three samples from northern Brazil, with a frequency of detection of 0.31%. Similar studies performed in northeast and south Brazil reported 0.13% and 0.36% frequencies of pestivirus detection in cattle, respectively [24,31]. The genetic identity was verified by sequencing the 5′UTR, N^pro^, and E2 coding regions, and all three were classified as HoBiPeV. Considering the key role of the northern region in the export of live cattle to other countries on different continents, it is critical to screen these animals using tests with increased sensitivity to HoBiPev, since the BVDV test may fail or have reduced sensitivity to detect HoBiPeV strains [60,61].

The HoBiPeV sequences analyzed in the present study are highly similar, supported by high bootstrap values, and clustered with several described sequences classified as subgenotype “a” [24,56]. The sequences described in Italy and Brazil are grouped into the same branch, but the sequence described in Asia, also previously classified as subgenotype “a”, is very divergent within this subgroup [62]. The origin and dispersion of HoBiPeV remain unclear, yet it probably emerged in Asia and subsequently spread to other regions around the world [27,63,64]. Previous studies with Bayesian analysis suggest a relatively recent origin for HoBiPeV [65,66]. The subgenotype “a” emerged more than a century after the initial putative Asian origin of the virus. These events coincide with an intense importation of water buffalos and cattle *Bos taurus indicus* from Asia to Brazil [63,64]. This hypothesis would explain the presence of HoBiPeV in regions raising significant water buffalo populations, such as Brazil and Asia [24,55]. Furthermore, animal movement between regions, sometimes with low sanitary control, could explain the high prevalence of HoBiPev in northern Brazil, similar to that described in the northeast region, suggesting the dissemination of the virus [67]. Until now, HoBiPeV presence in South America seems to be restricted to herds from Brazil [24,55,68] and Argentina [25].

In our study, the phylogeny of the 5′UTR for the sequence LV_510/20AM was not performed (Figure 2a), as it was not possible to amplify 5’UTF by RT-PCR using primers 324F and 326R. Analysis using the 5′UTR alone can be limited because of the short sequence length and low diversity. Phylogenetic analyses can be significantly improved by analyses of longer and more variable sequences and, therefore, it has been recommended to use the N^pro^ and E2 coding regions [69,70]. Therefore, pestivirus phylogeny should not be based on analysis of a single genomic fragment, which generates conflicting results for some isolates [71].

The prevalence of antibodies against pestiviruses in cattle herds worldwide is typically high. For instance, in Europe, Asia, and North America, antibody prevalence reaches 60 to 90% [72]. Similarly, serological studies in many Brazilian states show seropositivity rates ranging from 40 to 75% [73,74,75,76]. However, most of these studies performed serology using a single bovine pestivirus species. In our work, we performed a comparative VN test for BVDV-1, 2, and HoBiPeV in 390 bovine serum samples collected from northern Brazilian herds. The overall percentage of animals presenting antibody titers against pestiviruses was 46% (177/390). Of these 177 positive samples, 19.7%, 14%, and 20.8% were positive for BVDV-1, BVDV-2, and HoBiPeV, respectively. However, 45.2% (80/177) could not be determined. This may be related to the great genetic and antigenic diversity within the pestiviruses, associated with serologic cross-reaction between pestivirus species. Another explanation is that some of the non-classified samples were infected by more than one pestivirus species. Additionally, it cannot be excluded that these animals might be infected by one unknown circulating pestivirus.

The percentage of seropositive animals was similar between the species tested in the VN: BVDV-1, 2, and HoBiPeV. On the other hand, we only detected HoBiPev by RT–PCR. Interestingly, as in the present study, RT–PCR results in bovine serum samples from the northeast also only detected HoBiPeV [24].

VN tests performed with buffalo samples in Argentina showed that 32% of animals were seropositive for BVDV or HoBiPeV [77]. In bovine samples collected at the Texas/Mexico border, approximately 50% were seropositive for pestiviruses in the VN test [78]. In a USA seroprevalence study, more than 80% of bovine samples were seropositive for ruminant pestiviruses [79]. Comparing seroprevalence data among studies conducted in other regions of Brazil, it seems that BVDV is widely disseminated and there is a notable predominance of one pestivirus species over another varying between regions [73].

In Brazil, vaccination against BVDV-associated diseases is still incipient [73,80] and the herds analyzed in our study were most likely not vaccinated, not interfering with the VN results. In 2016, only 5 million BVDV-containing vaccine doses were sold, and most of these vaccines were in cattle herds in Brazil’s southern and central-western regions [81]. Most BVDV vaccines licensed in Brazil are inactivated and contain only BVDV-1a or BVDV-1a and BVDV-2b. Inactivated vaccines are safe, but can induce lower levels of antibody titers and generally require booster vaccinations [82]. Recently, two attenuated (modified live virus, MLV) vaccines have become available in Brazil. In contrast, MLV usually stimulates higher levels of antibody titers [83]. However, its use in pregnant females should be undertaken with caution, due to the inherent risk of transplacental infection [84]. To date, no commercial vaccines are available for HoBiPeV. Several vaccines are available in Brazil, but the high genetically and antigenically variability among pestiviruses results in low serologic reactivity between BVDV species and HoBiPeV [60,80,82]. Thus, there is a concern that the immunological response induced for commercial vaccines currently available in Brazil results in low levels of neutralization against HoBiPeV. It is necessary to improve the quality of vaccines available in Brazil. Including representative Brazilian strains of BVDV and HoBiPeV in new vaccine formulations will certainly contribute to better efficacy.

Serological cross-reactivity among pestivirus species and strains is variable, and several studies carried out with vaccines formulated with North American or European isolates showed poor production of antibody titers against Brazilian isolates [30,85]. The detection of high titers of pestiviruses in nonvaccinated herds may suggest the presence of PI animals [64]. Cattle that have serum titers lower than 16 are not considered protected against acute bovine pestivirus infection. To have the potential to confer fetal protection and prevent the birth of PI animals, titers must be higher than 256 [86,87,88]. Antibody titers against BVDV-1, BVDV-2, and HoBiPeV are shown in Appendix A. Thus, our results emphasize that the vaccines used in Brazil must be revised to include the pestivirus species and subgenotypes that predominate in the country.

## 5. Conclusions

This study provides the first evidence of the genetic and antigenic diversity of ruminant pestiviruses in the Brazilian Amazon region. The level of neutralizing antibodies evidences the circulation of BVDV-1, 2, and HoBiPeV. Our results showed 45.4% (177/390) of seropositive samples (19.8% for BVDV-1, 14.1% for BVDV-2, and 20.9% for HoBiPeV), with significant seroprevalence for HoBiPeV. In addition, the molecular results showed three positive samples classified as HoBiPeV clustered into subtype “a”, the same subgenotype as all HoBiPev sequences available from Brazil until now. Understanding the genetic and antigenic diversity of ruminant pestiviruses in northern Brazil, which plays an important role in national and international livestock cattle, is essential for better vaccine formulation, the use of accurate diagnostic tests for the detection of HoBiPev, and to support the planning of future intervention efforts for pestivirus control and eradication programs.

## Figures and Tables

**Figure 1 viruses-15-00453-f001:**
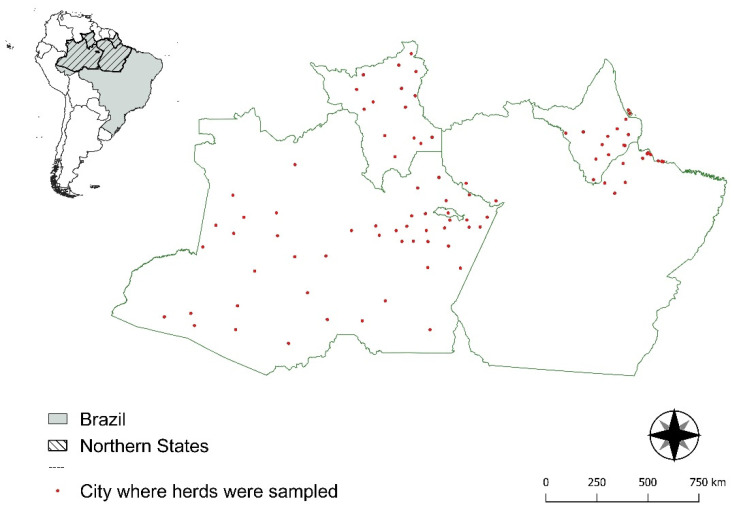
Spatial distribution of the cattle sampled to investigate the presence of ruminant pestiviruses in the states of Pará (PA), Amapá (AP), Roraima (RR), and Amazonas (AM), northern Brazil.

**Figure 2 viruses-15-00453-f002:**
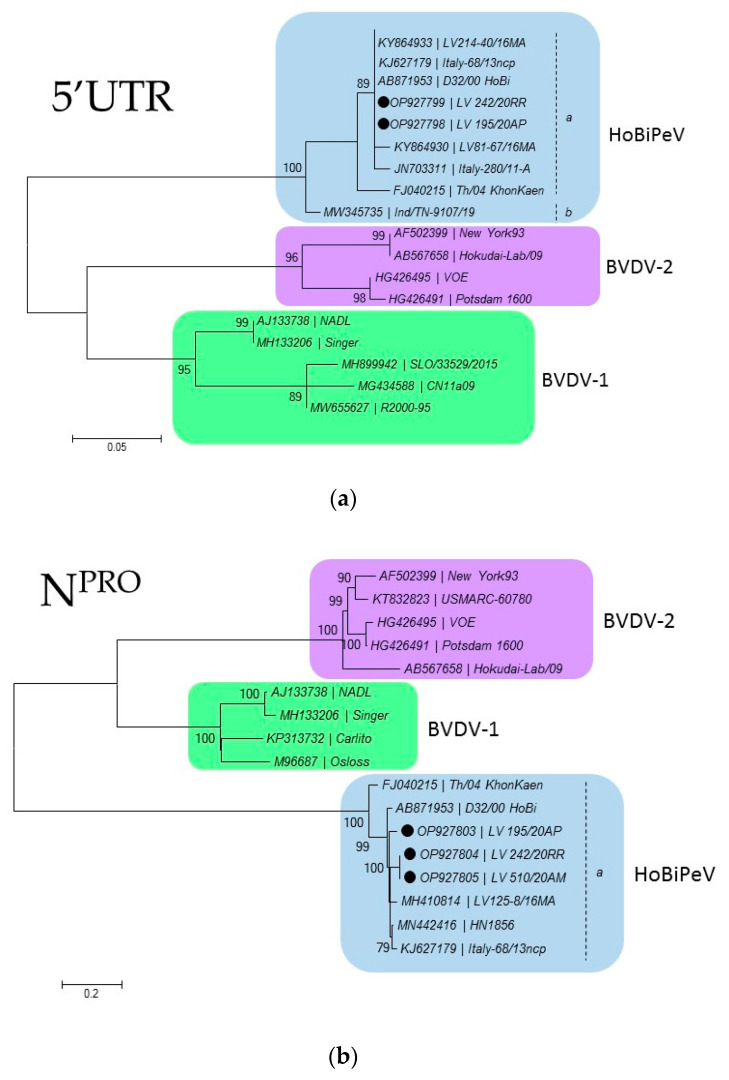
Molecular phylogenetic analysis by the maximum likelihood method. Phylogenetic tree based on the nucleotide sequence: (**a**) 5′UTR, (**b**) N^pro^, and (**c**) E2. Phylogenetic analysis for subgenotype classification was performed using the BVDV sequences obtained from GenBank. MEGA6 was used for phylogeny inference according to a maximum likelihood algorithm. Bootstrap analyses that were supported by >70% of 1000 replicates are indicated in nodes. The sequences from this study are highlighted with a ● symbol. The phylogeny of the 5′UTR for the sequence LV_510/20AM was not performed, since RT-PCR was negative, repetitively (Figure 2a).

**Figure 3 viruses-15-00453-f003:**
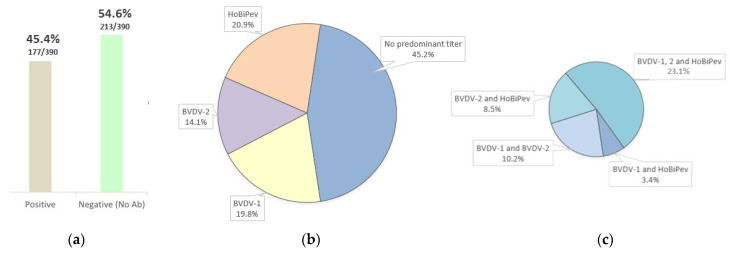
Neutralizing antibody titers were determined in bovine serum samples tested by VN assay against BVDV-1, BVDV-2, and HoBiPeV strains: (**a**) Percentage of samples with neutralizing antibodies against bovine pestiviruses; (**b**) During cross-neutralization, samples with titers higher than 4-fold for one specific species are considered to be raised by the infection of that species; (**c**) For cross-neutralization results with titers against more than one species (but <4-fold) the samples had no predominant titer against one specific pestivirus species.

**Table 1 viruses-15-00453-t001:** Description of the primers used for RT–PCR targeting the pestivirus 5′UTR, N^pro^, and E2.

Target	Primer	Pestivirus Species *	Reference
5′UTR	Pesti F; Pesti R	BVDV-1, BVDV-2	[40]
	324 F; 326 R **	BVDV-1, BVDV-2, HoBiPeV	[41]
	Panpesti F; Panpesti R	BVDV-1, BVDV-2, HoBiPeV	[31]
N^pro^	TF3; TR3 **	HoBiPeV	[42]
	LV Pesti F; LV PestiR	BVDV-2	[34]
	BD1, BD3	BVDV-1	[4]
E2	SF3, SR3 **	HoBiPeV	[43]
	E2F, E2R	BVDV-1, BVDV-2	[44]

* Ruminant pestivirus species detected according to the primer design and validation. ** Samples identified as positive when using these primers were submitted for Sanger sequencing and phylogenetic analysis.

**Table 2 viruses-15-00453-t002:** Number of samples for which the species that raised the antibodies could be determined (BVDV-1, 2 and HoBiPeV) or not, by federative state.

Federative State	VN Sample	Positive Samples			
		BVDV-1	BVDV-2	HoBiPeV	No Predominant Titer
AM	226	14	2	20	28
AP	57	5	5	9	13
PA	29	1	7	4	12
RR	78	15	11	4	27
**Total**	**390**	**35**	**25**	**37**	**80**

## Data Availability

Not applicable.

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
