# Peer review of "HoBi-like Pestivirus Is Highly Prevalent in Cattle Herds in the Amazon Region (Northern Brazil)"

_viruses, 2023, doi:10.3390/v15020453_

Round 1

Reviewer 1 Report

Viruses-2150278

HoBi-like pestiviruses were mainly described from Asia and South America (line 58: in Italy, it was not reported anymore since its first description in 2011; see Luzzago et al. (2021) Front Vet Sci 8, 669942). Brazil is one of the worldwide most important cattle producing country, but the diversity of bovine pestiviruses was obviously rarely analyzed in the Amazon region. As HoBi-like pestiviruses are quite divers compared to BVDV-I and –II, the use of vaccines and diagnostics tests need to be adapted to the presence of this pestivirus species and, the, the knowledge on the genetic diversity present in the cattle population is required to develop any control program. In this study Baumbach et al. investigate the prevalence of ruminant pestivirus in the Amazon region, using RT-PCR to detect viral RNA (mostly indicative of persistently infected animals), and VNT assay to detect neutralizing antibody and to differentiate antibodies to the various ruminant pestivirus species. Especially the latter assay is on the one hand rather labor-intensive, but on the other hand the only way to assign the antibodies to a given species, as no ELISA exist that could differentiate accordingly.

Specific  comments:

1) Line 39. For BVDV-1, up to subtype 1x (https://doi.org/10.3389/fvets.2022.1028866) was already described. Similarly, subgenotypes BVDV-2d and-2e were similarly proposed (https://doi.org/10.1007/s00705-022-05565-w). However, all these new subgenotypes are only proposed based on sequences in the 5'-UTR (not full ORF), and thus, this should be mentioned whether only full length subgenotypes will be reported or not.

2) Line 37: In pestivirus taxonomy, the term BVDV-3 should not be used anymore. In order to relate to the term in older literature, it can be mentioned that this term was formerly used.

3) Line 43: there are four structural proteins, not proteases. In addition, there are "at least" seven NS proteins, not exactly seven, as, e.g., NS2-3 is required for  virus assembly and NS3 is an essential part of the replication complex.

4) Line 72-79. When describing various numbers of cattle heads, it might be helpful for the reader to supply a table with the different regions in Brazil and the number of cattle heads, exports etc., e.g. as supplementary material. Currently, only a selection of numbers are provided, and an comparison between the different regions is difficult (e.g. "second largest Brazilian herd", but what is the largest?).

Similarly (line 230), "Brazil is the fifth largest exporter of live cattle globally and holds 3.6% of the world trade in live animals" sounds surprising, as 3.6 % is not too much for the 5th largest exporter, so what are the other four look like?

5) Chapter titles: Use either titles style with capital letters or not, but not mixed (e.g. "2.3 Sequencing and phylogenetic analysis" vs. "2.4 Virus Neutralization").

6) Line 151: Were the MDBK cells used tested freshly for BVDV contamination? Since 2015, some lots of CCL-22 from ATCC were contaminated with BVDV.

7) Line 169: It is not quite clear whether all 8 primer pairs were used in a pool and/or individually.

8) Results: It is advisable to provide a table (e.g. as supplementary material) providing ALL VNT data obtained. The global summary does not provide sufficient information in all cases, e.g. a less than fourfold difference might be obtained by low titers to all pestivirus species (e.g. 16 vs. 25), or also by high titers (e.g. 250 to BVDV-I and 390 to HoBi).

Similarly Figure 3b and 3c is unclear. Thus, "the highest titer (>4fold) is considered significant…", but 4-fold indicates a ratio between two titers, and similarly for 3c. "the highest titer <4 fold ???

9) Line 24 7ff: HoBi subgenotype are discussed, e.g. subgenotype "a", but the phylogenetic analyses in Fig. 2 are not sufficiently details to indicated the subgenotypes. If this is an important section of the discussion, this should be represented in the/a figure.

10) Line 262: Is there an explanation why the 5'-UTR of LV510/20AM could not be sequences? Thus appears rather surprising, as the 5'UTR is one of the most conserved regions of the pestiviral genome, and that's why it is often used in initial phylogenetic analysis.

11) Up to 45.2% of the antibody-positive sera could not be differentiated by cross-VNT (compare comment 8, providing all data). This value is rather high, and might indicate the that for Brazil (or at least this region in Brazil), not the optimal subgenotypes were chosen for the cross-VNT. It is not to be expected to repeat the VNT's with other strains, but the issue of subgenotypes used in the assay and the subgenotypes present in the area might be discussed in more detail.

12) Table S1: It does not make sense to have numbers with 2 decimal places, and others with none in the same table. In addition, provide the numbers (n) in the table.

13) The references need some editing, e.g. (list possibly not complete)
- Ref 46: Authors are Reed, L.J. and Muench, H., the volume is 27 (not £7), and is the link to the paper required? (https://academic.oup.com/aje/article/27/3/493/99616)
- Refs 65, 70, 72, 73: Provide full references, not just links.
- Ref 66, 75, 78. Provide also title in English, as done for ref. 68.
- Ref. 68; Complete reference, (it appears to be a master thesis), and here, it would make sense to provide the link.

Author Response

Thank you for volunteering your time to help increase the scientific value of our research. Thank you for your comments concerning our manuscript entitled “HoBi-like pestivirus is highly prevalent in cattle herds in the Amazon region (northern Brazil)”. We carefully have taken full account of all the reviewers’ suggestions. Those comments are very helpful for improving our paper. We are submitting the corrected and highlighted manuscript with the suggestions and corrections. The manuscript has been revised as per the comments given by the reviewers, and our responses to all the comments are as follows:

Response to Reviewer 1

Comments and Suggestions for Authors:

Point 1: HoBi-like pestiviruses were mainly described from Asia and South America (line 58: in Italy, it was not reported anymore since its first description in 2011; see Luzzago et al. (2021) Front Vet Sci 8, 669942).

Response 1: This part of the paragraph was rewritten with suggestions and added in the revised manuscript (Lines 79-84):” In recent decades, several studies have described the presence of this emerging pestivirus mainly in cattle from South America and southeast Asia. In 2011, HoBiPeV was described in an outbreak in southern Italy, however, until now, it has not been reported anymore in Europe.”

Specific comments:

1) Line 39: For BVDV-1, up to subtype 1x (https://doi.org/10.3389/fvets.2022.1028866) was already described. Similarly, subgenotypes BVDV-2d and-2e were similarly proposed (https://doi.org/10.1007/s00705-022-05565-w). However, all these new subgenotypes are only proposed based on sequences in the 5'-UTR (not full ORF), and thus, this should be mentioned whether only full length subgenotypes will be reported or not.

Response 1) Line 39: Thank you for pointing this out. This part of the paragraph was written based on ICTV information and data published in the literature. Since we do not mention the new subgenotypes based in the 5'UTR sequences alone, we did not state the criteria used here. We have added the information required in the revised manuscript (Lines 38-41): “Bovine pestiviruses can also be classified into subgenotypes based on phylogenetic analysis of full-length genome, 5'UTR sequences, or Npro and E2 analyses. To date, several subgenotypes have already been described: BVDV-1 (1a-1x), BVDV-2 (2a-2e), and HoBiPeV (a-e).”

2) Line 37: In pestivirus taxonomy, the term BVDV-3 should not be used anymore. In order to relate to the term in older literature, it can be mentioned that this term was formerly used.

Response 2) Line 37: We deleted BVDV-3 in that sentence. This change can be checked in line 37.

3) Line 43: there are four structural proteins, not proteases. In addition, there are "at least" seven NS proteins, not exactly seven, as, e.g., NS2-3 is required for virus assembly and NS3 is an essential part of the replication complex.

Response 3) Line 43: We apologize for the typing error in writing "proteases" instead of "proteins". This mistake has already been corrected and could be checked in Line 51.

We also included the term “at least” in Line 51.

4) Line 72-79: When describing various numbers of cattle heads, it might be helpful for the reader to supply a table with the different regions in Brazil and the number of cattle heads, exports etc., e.g. as supplementary material. Currently, only a selection of numbers are provided, and an comparison between the different regions is difficult (e.g. "second largest Brazilian herd", but what is the largest?).

Similarly (line 230), "Brazil is the fifth largest exporter of live cattle globally and holds 3.6% of the world trade in live animals" sounds surprising, as 3.6 % is not too much for the 5th largest exporter, so what are the other four look like?

Response 4) Line 72-79: Thank you for this suggestion. Information about herd size and exports has been updated with data for the year 2022. Also, a table with this information has been added to the supplementary material (S1 and S4) to help the reader compare this data in different regions of Brazil.

5) Chapter titles: Use either titles style with capital letters or not, but not mixed (e.g. "2.3 Sequencing and phylogenetic analysis" vs. "2.4 Virus Neutralization").

Response 5) Chapter titles: Thank you for your observation, the text was revised and rewritten according to your suggestions.

6) Line 151: Were the MDBK cells used tested freshly for BVDV contamination? Since 2015, some lots of CCL-22 from ATCC were contaminated with BVDV.

Response 6) Line 151: We understand, this is a concern for us too. Yes, all our MDBK passages are tested by RT-PCR, and result negative for BVDV.

7) Line 169: It is not quite clear whether all 8 primer pairs were used in a pool and/or individually.

Response 7) Line 169: Sorry for not describing it clearly. Positive pools were tested individually. This part of the paragraph was rephrased to reduce ambiguity in the revised manuscript (Line 196-199): “A total of 944 bovine serum samples were tested by RT‒PCR. Samples from positive pools were tested individually using the primer pairs listed in Table 1 in individual reactions. Three samples (0.31%) tested positive for pestivirus.”

8) Results: It is advisable to provide a table (e.g. as supplementary material) providing ALL VNT data obtained. The global summary does not provide sufficient information in all cases, e.g. a less than fourfold difference might be obtained by low titers to all pestivirus species (e.g. 16 vs. 25), or also by high titers (e.g. 250 to BVDV-I and 390 to HoBi).

Response: We agree with your suggestion that will increase the scientific value of the study. The table with all VNT results has been added as supplementary material (Table S3).

Similarly Figure 3b and 3c is unclear. Thus, "the highest titer (>4fold) is considered significant…", but 4-fold indicates a ratio between two titers, and similarly for 3c. "the highest titer <4 fold ??? titer higher thant ffolfd considered significant.

Response: We rephrased the sentence to make it clearer. (Line 245-253): “Figure 3. Neutralizing antibody titers were determined in bovine serum samples tested by VN assay against BVDV-1, BVDV-2, and HoBiPeV strains: (a) Percentage of samples with neutralizing antibodies against bovine pestiviruses; (b) During cross-neutralization, samples with titers higher than 4-fold for one specific species are considered to be raised by the infection of that species; (c) For cross-neutralization results with titers against more than one species (but <4-fold), the samples had no predominant titer against one specific pestivirus species.”

9) Line 247: HoBi subgenotype are discussed, e.g. subgenotype "a", but the phylogenetic analyses in Fig. 2 are not sufficiently details to indicated the subgenotypes. If this is an important section of the discussion, this should be represented in the/a figure.

Response 9) Line 247: Thank you for mentioning this important detail. We modified it in the revised manuscript (Lines 300-304) to show the different subgenotypes. “The HoBiPeV sequences analyzed in the present study are highly similar, supported by high bootstrap values, and clustered with several described sequences classified as subgenotype “a”. The sequences described in Italy and Brazil are grouped into the same branch, but the sequence described in Asia, also previously classified as subgenotype “a”, is very divergent within this subgroup.”

10) Line 262: Is there an explanation why the 5'-UTR of LV510/20AM could not be sequences? Thus appears rather surprising, as the 5'UTR is one of the most conserved regions of the pestiviral genome, and that's why it is often used in initial phylogenetic analysis.

Response 10) Line 262: It was really a surprise for us too. We have no explanation for not amplifying the 5’UTR of this sample using primers 324F and 326R. In the future, we are planning to sequence the whole genome of this strain. The sentence was rephrased in Lines 319-321: “In our study, the phylogeny of the 5'UTR for the sequence LV_510/20AM was not performed (Figure 2a), as it was not possible to amplify 5’UTF by RT-PCR using primers 324F and 326R.”

11) Up to 45.2% of the antibody-positive sera could not be differentiated by cross-VNT (compare comment 8, providing all data). This value is rather high, and might indicate the that for Brazil (or at least this region in Brazil), not the optimal subgenotypes were chosen for the cross-VNT. It is not to be expected to repeat the VNT's with other strains, but the issue of subgenotypes used in the assay and the subgenotypes present in the area might be discussed in more detail.

Response 11): Thank you for pointing this out! The choice of performing the VNT using the cytopathic (CP) strains BVDV-1a (Oregon C24 V), BVDV-2a (SV-253), and HoBiPeV (Italy 83/10) was based on similar previous studies that we wanted to compare with. These similar studies also had high rates of samples with no predominant titer (doi: 10.3389/fvets.2022.821247; doi: 10.1177/1040638716680251). Regarding the issue of subgenotypes present in the amazon region, there was no information on subgenotypes circulating in the region. Due to antigenic cross-reactivity between pestiviruses, many times is not possible to determine which species/subgenotype raise the highest titer or that the cattle was infected by one unknown pestivirus present in that region. We modified the text to emphasize this point in the revised manuscript (Lines: 336-341): “This may be related to the great genetic and antigenic diversity within the pestiviruses, associated with serologic cross-reaction between pestivirus species. Another explanation is that some of the non-classified samples were infected by more than one pestivirus species. Also, it cannot be excluded that these animals might be infected by one unknown circulating pestivirus.”

12) Table S1: It does not make sense to have numbers with 2 decimal places, and others with none in the same table. In addition, provide the numbers (n) in the table.

Response 12) Table S1: Thank you for the suggestion. The table was revised and can be checked in supplementary material (Table S1).

13) The references need some editing, e.g. (list possibly not complete)
- Ref 46: Authors are Reed, L.J. and Muench, H., the volume is 27 (not £7), and is the link to the paper required? (https://academic.oup.com/aje/article/27/3/493/99616)
- Refs 65, 70, 72, 73: Provide full references, not just links.
- Ref 66, 75, 78. Provide also title in English, as done for ref. 68.
- Ref. 68; Complete reference, (it appears to be a master thesis), and here, it would make sense to provide the link.

Response 13): The references were revised and rewritten according to suggestions. Thank you very much for your comments and suggestions to improve the manuscript.

OBSERVATION:

Dear Editor and Reviewers,

After suggestion, we have extended the content on the main text for the word count to >4000 words. The inclusions can be found at:

Lines 42-47

Lines 51-59

Lines 62-66

Lines 216-226

Lines 258-271

Lines 360-372

Thank you for your time and your comments concerning our manuscript.

Best Regards,

Cláudio Wageck Canal

Reviewer 2 Report

Baumbauch and colleagues have studied the distribution of pestiviruses in the Amazon region of Brazil. With a cattle population estimated at 52.4 million animals and a significant position in the international livestock market, the study is of visible epidemiological value - also because the sampling and scientific processing are coherent. Over 45% of the samples were seropositive for pestivirus antibodies in Brazil's low vaccination rate, and HoBiPeV was detected in three individual samples by RT-PCR. In comparative SNTs, reactivity against BVDV-1, BVDV-2, and HoBi-PeV was measured. Overall, the study is ready for publication if the DIscussion regarding the "unclear" reactants is improved (see Major points).

Minor points:

Line 43: Erns and not Erns

Line 47: BVDV and not Bovine viral diarrhea virus (BVDV)

Line 52: including and not including in

Line 177: Why are only two of the isolates seen in the 5'-UTR tree? Was there no successful RT-PCR for the one virus. Please mention as result and also comment in the legend.

Line 278: “However, 45.2% (80/177) could not be determined.” See: “Major points”

Major points:

1. If similar titers are obtained for the three virus species tested, it could be on one hand animals infected by more than one of the virus species. However, on the other, it could represent animals infected by a completely different pestivirus, so that only cross-reactivity becomes apparent to your viruses. Please discuss this result better. In light of the many new pestiviruses found in pigs (Bungowannah, Linda pestivirus, APPV), some of which can infect cattle cells, it would be useful to consider this possibility. 

2. A tabular listing of the individual virus titers in the form of an excel table would greatly enhance the scientific value of the study. This would also facilitate the processing of the data in later studies and improve the accessibility of the data.

Author Response

Thank you for volunteering your time to help increase the scientific value of our research. Thank you for your comments concerning our manuscript entitled “HoBi-like pestivirus is highly prevalent in cattle herds in the Amazon region (northern Brazil)”. We carefully have taken full account of all the reviewers’ suggestions. Those comments are very helpful for improving our paper. We are submitting the corrected and highlighted manuscript with the suggestions and corrections. The manuscript has been revised as per the comments given by the reviewers, and our responses to all the comments are as follows:

Response to Reviewer 2

Note: Some information about herd size and exports have been updated to year 2022.

Minor points:

Line 43: Erns and not Erns

Response (Line 51): We changed the word as pointed by the reviewer.

Line 47: BVDV and not Bovine viral diarrhea virus (BVDV)

Response (Line 56): Thank you, we changed the word.

Line 52: including and not including in

Response (Line 74): Thank you for the correction.

Line 177: Why are only two of the isolates seen in the 5'-UTR tree? Was there no successful RT-PCR for the one virus. Please mention as result and also comment in the legend.

Response: As described in the lines 319-321: In our study, the phylogeny of the 5'UTR for the sequence LV_510/20AM was not performed (Figure 2a), as it was not possible to amplify 5’UTF by RT-PCR using primers 324F and 326R. It was really a surprise for us, as the 5'UTR is one of the most conserved regions of the pestivirus genome. In the future, we are planning to sequence the whole genome of this sample. We appreciate your suggestion and tried to turn it clear in Results (lines 205-208) and legend of the figure (lines 198-200). The sentence was rephrased in Lines 284-285: “The phylogeny of the 5'UTR for the sequence LV_510/20AM was not performed (Figure 2a), since RT-PCR resulted negative, repetitively.”

Line 278: “However, 45.2% (80/177) could not be determined.” See: “Major points”

Major points:

  1. If similar titers are obtained for the three virus species tested, it could be on one hand animals infected by more than one of the virus species. However, on the other, it could represent animals infected by a completely different pestivirus, so that only cross-reactivity becomes apparent to your viruses. Please discuss this result better. In light of the many new pestiviruses found in pigs (Bungowannah, Linda pestivirus, APPV), some of which can infect cattle cells, it would be useful to consider this possibility.

Response 1: Thank you so much for your precious suggestion. Due to antigenic cross-reactivity between pestiviruses, many times is not possible to determine which species has the highest titer. In that study, 45.2% of the samples could not be determine had a higher titer against one of the pestivirus. Similar studies also had high rates of samples with no predominant titer (doi: 10.3389/fvets.2022.821247; doi: 10.1177/1040638716680251). We have modified de text to emphasize this point in the revised manuscript (Lines: 337-342): “This may be related to the great genetic and antigenic diversity within the pestiviruses, associated with serologic cross-reaction between pestivirus species. Another explanation is that some of the non-classified samples were infected by more than one pestivirus species. Also, it cannot be excluded that these animals might be infected by one unknown circulating pestivirus.”

  1. A tabular listing of the individual virus titers in the form of an excel table would greatly enhance the scientific value of the study. This would also facilitate the processing of the data in later studies and improve the accessibility of the data.

Response 2: One table with all VNT results has been added to the supplementary material (Table S3). We thank you very much for your comments and suggestions for improving our research.

OBSERVATION:

Dear Editor and Reviewers,

After suggestion, we have extended the content on the main text for the word count to >4000 words. The inclusions can be found at:

Lines 42-47

Lines 51-59

Lines 62-66

Lines 216-226

Lines 258-271

Lines 360-372

Thank you for your time and your comments concerning our manuscript.

Best Regards,

Cláudio Wageck Canal
